# Off-Policy Interval Estimation with Lipschitz Value Iteration

**Ziyang Tang**
University of Texas at Austin
ztang@utexas.edu

**Yihao Feng**
University of Texas at Austin
yihao@cs.utexas.edu

**Na Zhang**
Tsinghua University
zhangna@pbcsf.tsinghua.edu.cn

**Jian Peng**
University of Illinois at Urbana-Champaign
jianpeng@illinois.edu

**Qiang Liu**
University of Texas at Austin
lqiang@cs.utexas.edu

## Abstract

Off-policy evaluation provides an essential tool for evaluating the effects of different policies or treatments using only observed data. When applied to high-stakes scenarios such as medical diagnosis or financial decision-making, it is crucial to provide provably correct upper and lower bounds of the expected reward, not just a classical single point estimate, to the end-users, as executing a poor policy can be very costly. In this work, we propose a provably correct method for obtaining interval bounds for off-policy evaluation in a general continuous setting. The idea is to search for the maximum and minimum values of the expected reward among all the Lipschitz Q-functions that are consistent with the observations, which amounts to solving a constrained optimization problem on a Lipschitz function space. We go on to introduce a Lipschitz value iteration method to monotonically tighten the interval, which is simple yet efficient and provably convergent. We demonstrate the practical efficiency of our method on a range of benchmarks.

## 1 Introduction

Reinforcement learning (RL) (e.g., Sutton & Barto, 1998) has become widely used in tasks like recommendation system, robotics, trading and healthcare (Murphy et al., 2001; Li et al., 2011; Bottou et al., 2013; Thomas et al., 2017). The current success of RL highly relies on excessive amount of data, which, however, is usually not available in many real world tasks where deploying a new policy is very costly or even risky. Off-policy evaluation (OPE) (e.g., Fonteneau et al., 2013; Jiang & Li, 2016; Liu et al., 2018a), estimating the expected reward of a target policy using observational data gathered from previous behavior policies, therefore holds tremendous promise for designing data-efficient RL algorithms by leveraging on previously collected data.

Existing OPE methods mainly focus on *point estimation*, which only provides a single point estimation of the expected reward. However, such point estimate can be rather unreliable as OPE often suffers from high error due to the lack of historical samples, policy shift or model misspecification. Further, for applications in high-stakes areas such as medical diagnosis and financial investment, a point estimate itself is far from enough and can even be dangerous if it is unreliable. Hence, it is essential

to provide *provably correct interval estimation* of the expected reward, which is both trustful and theoretically correct.

To address this problem, we propose a general optimization-based framework to derive a provably correct off-policy interval estimation based on historical samples. Our idea is to search for the largest and smallest possible values of the expected reward, among all the Q-functions in a Lipschitz function space that are consistent with the observed historical samples. This interval estimator is provably correct once the true Q-function satisfies the Lipschitz assumption.

Computing our upper and lower bounds amounts to solving a constrained optimization problem in the space of Lipschitz functions. We introduce a *Lipschitz value iteration* algorithm of a similar style to value iteration. Our method is efficient and provably convergent. In particular, our algorithm has a simple closed form update at each iteration and is guaranteed to monotonically tighten the bounds with a linear rate under mild conditions. To speed up the algorithm, we develop a double subsampling strategy, which we only pick a random subsample of value functions to update in each iteration and use the same batch of data as constraints.

We test our algorithm on a number of benchmarks and show that it can provide tight and provably correct bounds.

**Related Work**    Our work is closely related to the off-policy point estimation. There are typically two types of OPE methods, importance sampling (IS) based methods (e.g., Liu, 2001; Precup et al., 2000; Liu et al., 2018a; Xie et al., 2019) and value function or model based methods (e.g., Fonteneau et al., 2013; Liu et al., 2018b; Le et al., 2019; Feng et al., 2019) Another line of work combines these two methods and yields a doubly-robust estimator for off-policy evaluation (Jiang & Li, 2016; Thomas & Brunskill, 2016; Kallus & Uehara, 2019; Tang et al., 2020). In this work, we consider the black box setting when the behavior policy is assumed to be unknown (e.g., Nachum et al., 2019; Mousavi et al., 2020; Zhang et al., 2020; Feng et al., 2020).

IS-based point estimation methods naturally yield a confidence interval by standard concentration (Thomas et al., 2015b,a). Another major type of confidence interval estimation approaches leverages the statistical procedure of bootstrapping (White & White, 2010; Hanna et al., 2017). However, these confidence intervals are typically loose due to the curse of horizon. Moreover, they heavily rely on the assumption that the off-policy data is drawn i.i.d. from a particular behavior policy. But this is not always true since the policies usually evolve and depend on their previous policies.

Another related set of works are PAC-RL (e.g., Jin et al., 2018; Dann et al., 2018; Song & Sun, 2019; Yang et al., 2019), which mainly focus on the regret bound or sample complexity of the Q-learning exploration. As a side product, they also provide a confidence interval estimation for the value function. However, this line of work mostly focuses on the tabular or linear MDPs. In contrast, our work aims to handle the general continuous MDPs by leveraging Lipschitz properties of Q-functions. Song & Sun (2019) provides a metric embedding style Q-learning method, with a particular focus on finite horizon settings.

## 2   Background and Problem Settings

We firstly set up the problem of off-policy interval evaluation and then introduce the related background on Q-learning and Bellman equation.

**Markov Decision Process**    A Markov decision process (MDP) $M = \langle \mathcal{S}, \mathcal{A}, r, \boldsymbol{T}, \mu_0, \gamma \rangle$ consists of a state space $\mathcal{S}$, an action space $\mathcal{A}$, an unknown deterministic reward function $r : \mathcal{S} \times \mathcal{A} \to \mathbb{R}$, an unknown transition function $\boldsymbol{T} : \mathcal{S} \times \mathcal{A} \to \mathcal{S}$, an initial state distribution $\mu_0$, and a discounted factor $\gamma$. Throughout this work, we assume that the transition and reward function are deterministic for simplicity and we can easily draw samples from the initial state distribution $\mu_0$.

In RL, an agent acts in a MDP following a policy $\pi(\cdot|s)$, which prescribes a distribution over the action space $\mathcal{A}$ given each state $s \in \mathcal{S}$. Running the policy starting from the initial distribution $\mu_0$ yields a random trajectory $\tau := \{s_i, a_i, r_i\}_{1 \le i \le T}$, where $s_i, a_i, r_i$ represent the state, action, reward at time $i$ respectively. We define the infinite horizon discounted reward of $\pi$ as $R^\pi :=$ $\lim_{T \to \infty} \mathbb{E}_{\tau \sim \pi} \left[ \sum_{i=0}^{T} \gamma^i r_i \right]$ , where $\gamma \in (0, 1)$ is a discounting factor; $T$ is the horizon length of the trajectory, which we assume to approach infinite, hence yielding an *infinite horizon problem*; $\mathbb{E}_{\tau \sim \pi}[\cdot]$ denotes the expectation of the random trajectories collected under the policy $\pi$.

**Black Box Off-Policy Interval Estimation** We are interested in the problem of *black-box off-policy interval evaluation*, which requires arguably the minimum assumptions on the off-policy data. It amounts to providing an interval estimation $[\underline{R^\pi}, \overline{R^\pi}]$ of the expected reward $R^\pi$ of a policy $\pi$ (called the target policy), given a set of transition pairs $\{s_i, a_i, s_i', r_i\}_{i=1}^n$ collected under a *different, unknown* behavior policy, or even a mix of different policies; here $s_i' = \boldsymbol{T}(s_i, a_i)$ and $r_i = r(s_i, a_i)$ denote the next state and the local reward following $(s_i, a_i)$ respectively.

**Q-function and Bellman Equation** We review the properties of the Q-function that is most relevant to our work. The Q-function $Q^\pi(s, a)$ specifies the expected reward when following $\pi$ from the state-action pair $(s, a)$ and is known to be the unique fixed point of the Bellman equation:

$$Q(s, a) = \mathcal{B}^\pi Q(s, a) := r(s, a) + \gamma \mathcal{P}^\pi Q(s, a), \quad \forall (s, a) \in \mathcal{S} \times \mathcal{A}, \tag{1}$$

where $\mathcal{B}^\pi$ denotes the Bellman operator, and $\mathcal{P}^\pi$ is the transition operator defined by

$$\mathcal{P}^\pi Q(s, a) := \mathbb{E}_{s' = \boldsymbol{T}(s,a), a' \sim \pi(\cdot | s')} [Q(s', a')]. \tag{2}$$

An expected reward associated with a Q-function is defined via

$$R_{\mu_0, \pi}[Q] := \mathbb{E}_{s_0, a_0 \sim \mu_{0,\pi}} [Q(s_0, a_0)], \tag{3}$$

where we use $\mu_{0,\pi}(s, a) = \mu_0(s)\pi(a|s)$ to denote the joint initial state-action distribution.

**Off-Policy Q-Learning** Q-function can be learned in an off-policy manner. Assume we have a set of transition pairs $\mathcal{D} := \{s_i, a_i, s_i, r_i'\}_{i=1}^n$. Under the assumption of deterministic transition and reward, we can estimate the Bellman operator on each of the data point:

$$\mathcal{B}^\pi Q(s_i, a_i) = r_i + \mathbb{E}_{a' \sim \pi(\cdot | s_i')} [Q(s_i', a')],$$

which can be estimated with an arbitrarily high accuracy via drawing a large number of samples from $\pi(\cdot | s_i')$. Assume $Q^\pi$ belongs to a class of functions $\mathcal{F}$, we can estimate $Q^\pi$ by finding a $Q \in \mathcal{F}$ that satisfies the Bellman equation on the data points:

$$Q \in \mathcal{F}, \quad s.t. \quad Q(s_i, a_i) = \mathcal{B}^\pi Q(s_i, a_i), \quad \forall i \in [n]. \tag{4}$$

Compared with the exact Bellman equation (1), we only match the equation on the observed data (4), which may yield multiple or even infinite solutions of $Q$. Therefore, the function class $\mathcal{F}$ needs to be sufficiently constrained in order to yield meaningful solutions. In practice, (4) is often solved using fitted value iteration (Munos & Szepesvári, 2008), which starts from an initial $Q_0$, and then perform iterative updates by

$$Q_{t+1} \leftarrow \arg\min_{Q \in \mathcal{F}} \sum_{i=1}^n (Q(s_i, a_i) - \mathcal{B}^\pi Q_t(s_i, a_i))^2. \tag{5}$$

It is easy to see $R^\pi = R[Q^\pi]$. With an estimation of $Q^\pi$, the expected reward $R^\pi$ in (3) can be estimated by Monte Carlo sampling from $\mu_0$ and $\pi$.

## 3 Main Method

We now introduce our main framework for providing provably correct upper and lower bounds of the expected reward. For notation, we use $x = (s, a)$ to represent a state-action pair.

### 3.1 Motivation and Optimization Framework

When the fitted value iteration in (5) converges, it only provides one Q-function that yields a point estimation of the reward. To get an interval estimation, we expect the fitted value iteration to provide two Q-functions $\overline{Q}$ and $\underline{Q}$, such that all possible $Q$ consistent with the data points lie between $\overline{Q}$ and $\underline{Q}$, e.g. $\overline{Q} \succeq Q \succeq \underline{Q}$, where $f \succeq g$ means $f(x) \geq g(x), \forall x$.

More concretely, consider a function set $\mathcal{F}$ that is expected to include $Q^\pi$, we construct an upper bound of $R^\pi$ by

$$\overline{R_{\mathcal{F}}^\pi} = \sup_{Q \in \mathcal{F}} \left\{ R_{\mu_0, \pi}[Q], \quad s.t. \quad Q(x_i) \leq \mathcal{B}^\pi Q(x_i), \quad \forall i \in [n] \right\}, \tag{6}$$

where $R[Q]$ is defined in (3). It is easy to see that $\overline{R^\pi_{\mathcal{F}}} \geq R^\pi$ if $Q^\pi \in \mathcal{F}$ holds; this is because $Q^\pi$ satisfies the constraints in the optimization, and hence $\overline{R^\pi_{\mathcal{F}}} \geq R[Q^\pi] = R^\pi$ as a result of the optimization.

It is worthy to note that we use the Bellman inequality constraint in (6), which would not cause any looseness compared to the equality constraint. To see this, note that the exact reward $R^\pi$ can be framed into (Bertsekas, 2000)

$$R^\pi = \sup_{Q \in \mathcal{F}} \left\{ R_{\mu_0,\pi}[Q], \ \text{s.t.} \ Q(x) \leq \mathcal{B}^\pi Q(x), \ \forall x \in \mathcal{S} \times \mathcal{A} \right\},$$

which implies that $\overline{R^\pi_{\mathcal{F}}}$ would converge to $R^\pi$ as data points increase.

We can construct the lower bound in a similar way:

$$\underline{R^\pi_{\mathcal{F}}} = \inf_{Q \in \mathcal{F}} \left\{ R_{\mu_0,\pi}[Q], \ \text{s.t.} \ Q(x_i) \geq \mathcal{B}^\pi Q(x_i), \ \forall i \in [n] \right\}. \tag{7}$$

Define $I_{\mathcal{F}} = [\underline{R^\pi_{\mathcal{F}}}, \overline{R^\pi_{\mathcal{F}}}]$ as the interval estimation for $R^\pi$, once the true $Q^\pi$ lies in the function space $\mathcal{F}$, $R^\pi$ lies in $I_{\mathcal{F}}$ provably.

**Benefits of Our Framework**   Our optimization framework enjoys several advantages compared with the existing methods. First of all, unlike the standard concentration bounds, our bounds do not rely on the i.i.d. assumption of transition pairs $\{s_i, a_i\}$. In RL settings, the historical transition pairs are highly dependent to each other: on one hand, in sequential data stream, the current step of state is the next state in the previous step; on the other hand, the behavior policy that generates the trajectories is also evolving during the learning process. Secondly, under our optimization framework, more data would enable us to add more constraints in our searching space and therefore get tighter bounds accordingly. This property allows a trade-off between the time complexity and the tightness of the bounds, which we will further discuss in section 4.1. Last but not least, the tightness of the bounds depends on the capacity of the function space $\mathcal{F}$, which naturally yields the nice monotonicity property. It is easy to see if $Q^\pi \in \mathcal{F}_1 \subset \mathcal{F}_2$, we will have $R^\pi \in I_{\mathcal{F}_1} \subset I_{\mathcal{F}_2}$.

**Lipschitz Function Space**   To implement this framework, it is important to choose a proper function set $\mathcal{F}$ and solve the optimization process efficiently. Intuitively, $\mathcal{F}$ should be rich enough to include the true value function, but not be too large to cause the bounds to be vacuous. In light of this, we propose to use the space of Lipschitz functions. Let $\mathcal{X}$ be a metric space equipped with a distance $d\colon \mathcal{X} \times \mathcal{X} \to \mathbb{R}$. We propose to take $\mathcal{F}$ to be a ball in the Lipschitz function space:

$$\mathcal{F}_\eta := \{f\colon \|f\|_{d,\text{Lip}} \leq \eta\}, \qquad \text{where} \qquad \|f\|_{d,\text{Lip}} = \sup_{x,x' \in \mathcal{X}, \ x \neq x'} \frac{|f(x) - f(x')|}{d(x, x')}, \tag{8}$$

where $\|f\|_{d,\text{Lip}}$ is the Lipschitz norm of $f$ and $\eta$ is a radius parameter.

Although the Lipschitz space yields an infinite dimensional optimization, our key technical contribution is to show that it is possible to calculate the upper and lower bounds with an efficient fixed point algorithm. We form it as an optimization problem and solve it efficiently in a value iteration like fashion. In addition, in Section 3.4, we show that the size of the Lipschitz space enables us to establish a diminishing bound on the gap of the upper and lower bounds.

## 3.2   Optimization in Lipschitz Function Space

We propose a novel value iteration-like algorithm for solving the optimizations in (6) and (7). We focus on the upper bound (6), as the case of the lower bound is similar and discussed in Appendix A.

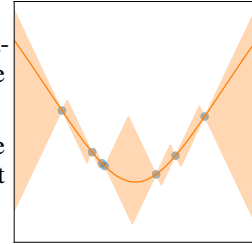

Our algorithm enjoys the same spirit as the fitted value iteration (5). We apply the Bellman operator on the current estimation of $Q$, and then use it as the constraint on the Bellman inequalities to obtain a new estimation.

Specifically, starting from an initial $\overline{Q}_0$ with $\overline{q}_{0,i} := \overline{Q}_0(x_i)$, at the $t$-th iteration, we update by

Figure 1: Illustration

$$\overline{Q}_t = \arg\max_{Q \in \mathcal{F}} \left\{ R_{\mu_0,\pi}[Q], \ \text{s.t.} \ Q(x_i) \leq \overline{q}_{t,i}, \forall i \in [n] \right\}, \quad \overline{q}_{t+1,i} = \mathcal{B}^\pi \overline{Q}_t(x_i), \quad \forall i \in [n]. \tag{9}$$

This update requires to solve a constrained optimization on the space $\mathcal{F}$. For our choice of the Lipschitz function space, this yields a simple closed form solution.

**Algorithm 1** Lipschitz Value Iteration (for Upper Bound)

---

**Input**: Transition data $\mathcal{D} = \{s_i, a_i, s'_i, r_i\}_{1 \le i \le n}$; discounted factor $\gamma$; target policy $\pi$; Lipschitz constant $\eta$.
**Initialize** $\overline{q}_{0,i}$ with criterion like (12).
**for** iteration $t$ **do**
    **Update:** $\overline{q}_{t+1,i} = \mathcal{B}^\pi \overline{Q}_t(x_i)$,    where $\overline{Q}_t(x) = \min_{j \in [n]} \{\overline{q}_{t,j} + \eta d(x, x_j)\}$
**end for**
**Return**: upper bound: $\overline{R_{\mathcal{F}}^\pi} = \mathbb{E}_{s,a \sim \mu_{0,\pi}} \left[ \min_{j \in [n]} \{\overline{q}_{t,j} + \eta d(x, x_j)\} \right]$.

---

**Proposition 3.1.** *Suppose $\mu_{0,\pi}(x) > 0$, $\forall x \in \mathcal{S} \times \mathcal{A}$, consider the optimization in (9) with $\mathcal{F} = \mathcal{F}_\eta$. We have*

$$\overline{Q}_t(x) = \min_{j \in [n]} (\overline{q}_{t,j} + \eta d(x, x_j)), \qquad \overline{q}_{t+1,i} = \mathcal{B}^\pi \overline{Q}_t(x_i), \quad \forall i \in [n]. \tag{10}$$

Intuitively, $\overline{Q}_t$ in (10) yields an

*upper envelope* of all the possible Lipschitz functions $Q$ that satisfy $Q(x_i) \le \overline{q}_{t,i}$, $\forall i$, and hence solves the optimization in (9), as $R[Q]$ is monotonically increasing with $Q$. The updates for the lower bound can be derived similarly by calculating the *lower envelopes*:

$$\underline{Q}_t(x) = \max_{i \in [n]} (\underline{q}_{t,i} - \eta d(x, x_i)), \qquad \underline{q}_{t+1,i} = \mathcal{B}^\pi \underline{Q}_t(x_i), \forall i \in [n]. \tag{11}$$

Note that these updates can be simplified to only keeping track of the Q-function values at the data points $\{\overline{q}_{t,i}\}_{i=1}^n$, as summarized in Algorithm 1. Figure 1 illustrates how the upper and lower envelopes bound all the possible Lipschitz functions going through the same set of data points.

### 3.3 Convergence Analysis

Our algorithm enjoys two important and desirable properties: one is monotonic convergence, which indicates that you can stop any time, depending on your time budget, and still get a valid lower/upper bound; the other is linear convergence, which implies that it only needs logarithmic time steps to converge. Thanks to these properties, our method is much faster compared to directly solving the convex optimization in (6) with (stochastic) sub-gradient ascent.

The monotonic convergence relies on the initial upper bound we pick, which is inspired by the monotonicity of the Bellman operator. See Appendix B.1 for more details.

**Theorem 3.2.** *Following the update in* (10) *starting from*

$$\overline{q}_{0,i} = \frac{1}{1 - \gamma} \left( r_i + \gamma \eta \mathbb{E}_{x'_i \sim \boldsymbol{T}^\pi(\cdot | x_i)} [d(x_i, x'_i)] \right), \tag{12}$$

*we have*

$$\overline{Q}_t \succeq \overline{Q}_{t+1} \succeq \overline{Q}^\pi, \ \forall t = 0, 1, 2, \dots, \tag{13}$$

*where $\overline{Q}^\pi = \arg \max_{Q \in \mathcal{F}} \{R[Q], \ s.t. \ Q(x_i) \le \mathcal{B}^\pi Q(x_i), \ \forall i \in [n]\}$. Therefore,*

$$R_{\mu_{0,\pi}}[\overline{Q}_t] \ge R_{\mu_{0,\pi}}[\overline{Q}_{t+1}] \ge \overline{R_{\mathcal{F}}^\pi} \ge R^\pi,$$

*and $\lim_{t \to \infty} R_{\mu_{0,\pi}}[\overline{Q}_t] = \overline{R_{\mathcal{F}}^\pi}$.*

We establish a fast linear convergence rate for the updates in (10). The convergence result here does not need the initialization of $\overline{q}_0$.

**Proposition 3.3.** *Following the updates in* (10) *under arbitrary initialization, with constant $C := \max_{i \in [n]} |\overline{q}_{1,i} - \overline{q}_{0,i}|$ we have*

$$R_{\mu_{0,\pi}}[\overline{Q}_t] - \overline{R_{\mathcal{F}}^\pi} \le C \frac{\gamma^t}{1 - \gamma}.$$

---

**Algorithm 2** Lipschitz Value (Upper Bound) Iteration with Stochastic Update

---

**Input**: Transition data $\mathcal{D} = \{s_i, a_i, s_i', r_i\}_{1 \leq i \leq n}$; discounted factor $\gamma$; target policy $\pi$; distance function $d$. Lipschitz constant $\eta$. Subsample size $n_B$.

**Initialize**: $\overline{q}_{0,i} = \frac{1}{1-\gamma}(r_i + \gamma\eta\widehat{\mathbb{E}}_{x \sim \boldsymbol{T}^\pi(\cdot|x_i)}[d(x_i, x)])$, $\forall i$, according to equation (12).

**for** iteration t **do**

    Subsample a subset $S_t \subseteq \{1, 2, ..., n\}$ with $|S_t| = n_B$.

    **Update**: $\overline{q}_{t+1,i} = \min\{\overline{q}_{t,i}, \mathcal{B}^\pi \overline{Q}_t(x_i)\}$, $\forall i \in S_t$ and $\overline{q}_{t+1,i} = \overline{q}_{t,i}$, $\forall i \notin S_t$ where $\overline{Q}_t(x) = \min_{j \in S_t}\{\overline{q}_{t,j} + \eta d(x_j, x)\}$.

**end for**

**Return**: upper bound: $\overline{R_\mathcal{F}^\pi} = \widehat{\mathbb{E}}_{x \sim \mu_{0,\pi}}[\min_{i \in [n]}\{q_{T,i} + \eta d(x_i, x)\}]$.

---

## 3.4 Tightness of Lipschitz-Based Bounds

We provide a quantitative estimation of the gap $(\overline{R_\mathcal{F}^\pi} - \underline{R_\mathcal{F}^\pi})$ between the upper and lower bounds when using the Lipschitz function set. We show that it depends on a notion of the covering radius of the data points in the domain.

**Theorem 3.4.** *Let $\mathcal{F} = \mathcal{F}_\eta$ be the Lipschitz function class with Lipschitz constant $\eta$. Suppose $\mathcal{X} = \mathcal{S} \times \mathcal{A}$ is a compact and bounded domain equipped with a distance $d \colon \mathcal{X} \times \mathcal{X} \to \mathbb{R}$. For a set of data points $X = \{x_i\}_{i=1}^n$, we have*

$$\overline{R_\mathcal{F}^\pi} - \underline{R_\mathcal{F}^\pi} \leq \frac{2\eta}{1-\gamma}\varepsilon_X,$$

*where $\gamma$ is the discount factor and $\varepsilon_X = \sup_{x \in \mathcal{X}} \min_i d(x, x_i)$ is the covering radius.*

Typically, the covering radius in a bounded and compact domain asymptotically grows with $O(n^{-1/\tilde{d}})$, where $\tilde{d}$ is an intrinsic dimension of the domain (Cohen et al., 1985), if the support of the data distribution (where $x_i$ is getting from) covers $d_\pi$, the stationary distribution of policy $\pi$. This shows that the bound gets tighter as the number of samples get larger, but may decay slowly when the data domain has a very high intrinsic dimension. While it is possible to choose smaller space sets (such as RKHS) to obtain smaller gaps, it would sacrifice other properties such as capacity and simplicity.

# 4 Practical Considerations

We discuss some practical concerns of Lipschitz value iteration in this section.

## 4.1 Accelerating with Stochastic Subsampling

If we draw $D$ samples to estimate $\mathcal{B}^\pi$ when updating with equation (10), we need $O(n^2 D)$ times of calculations for each round. This $n^2$ comes from two sources, one is the updating of all $n$ terms of $q_{t,i}$ in each iteration, and the other is taking the maximum/minimum among all $n$ upper/lower curves when we calculate the upper/lower envelope function of $\overline{Q}_t(x)$. This is quadratic to the number of samples and therefore computationally expensive as the sample size grows large.

To tackle this problem, we propose a fast random subsampling technique. Instead of updating all $n$ $q_{t,i}$ in each iteration, we pick a batch of subsamples $\{\overline{q}_{t,i}\}_{i \in S}$ to update, where $S \in \{1, 2, \ldots, n\}$ is a subset with fix size $|S| = n_B$. The benefit of subsampling is that we can trade-off between time-complexity and tightness.

To be more precise, we can write down the new update scheme as follow:

$$\overline{Q}_t(x) = \min_{j \in S_t}(\overline{q}_{t,j} + \eta d(x, x_j)),$$

$$\overline{q}_{t+1,i} = \min\{\overline{q}_{t,i}, \mathcal{B}^\pi \overline{Q}_t(x_i), \forall i \in S_t, \quad \overline{q}_{t+1,i} = \overline{q}_{t,i}, \forall i \notin S_t\}, \tag{14}$$

where $S_t$ is the sub-sample set in the $t$ iteration. A similar monotonic result can be shown in sequel.

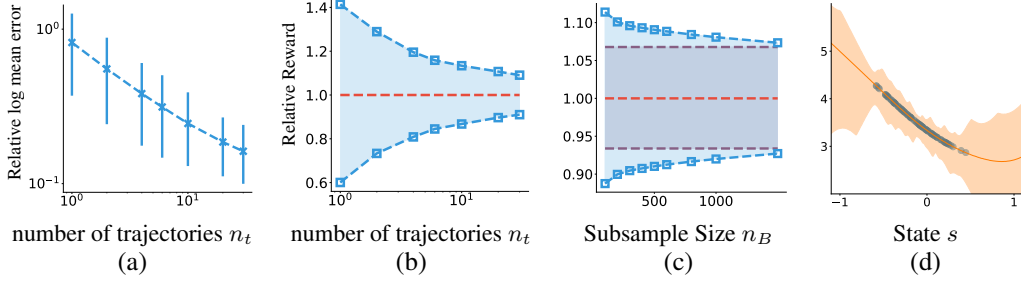

Figure 2: Results for synthesis environment with a *known* value function. The default settings: number of trajectory $n_t = 30$, Horizon length $H = 100$, discounted factor $\gamma = 0.95$, Lipschitz constant $\eta = 2.0$ and subsample size $n_B = 500$. (a) y-axis: log of relative mean error $\log((\overline{R_{\mathcal{F}}^\pi} - \underline{R_{\mathcal{F}}^\pi})/R^\pi)$; (b)(c) y-axis: relative reward; (d) landscape for value function $V^\pi$ with $[\overline{V^\pi}, \underline{V^\pi}]$, here state is bounded in interval $[-1.2, 1.2]$, interval is estimated using 100 samples. Blue curves: subsamples bounds; purple lines: the whole samples bounds.

**Proposition 4.1.** *Consider the update in* (14) *with initialization following* (12)*, let $\overline{Q}_t$ be the upper envelope function of data points $\{x_i, \overline{q}_{t,i}\}_{i=1}^n$, we have a similar monotonic result as theorem 3.2:*

$$\overline{Q}_t \succeq \overline{Q}_{t+1} \succeq \overline{Q}^\pi, \ \forall t = 0, 1, 2, \dots.$$

We summarize the strategy in Algorithm 2.

The subsampling algorithm only needs $O(n_B^2 D)$ time complexity in each iteration. Although this strategy does not converge to the exact optimal solution $\overline{R_{\mathcal{F}}^\pi}$, it still gives valid (despite less tight) bounds. In the numerical experiments, we find that once the size of subsample set is sufficiently large, we get almost the same tightness as the exact optimal bounds.

## 4.2 On Estimating Lipschitz Norm

The only model assumption we need to specify is the function set $\mathcal{F}_\eta$; we want $\eta \geq \|Q^\pi\|_{d,\mathrm{Lip}}$, but not to be too large, since the estimated interval gets loose as $\eta$ increases. However, compared to other value-based function approximation methods that also need assumptions on model specification, our non-parametric Lipschitz assumption is obviously very mild.

In order to set hyperparameter $\eta$, we would like to estimate the upper bound of $\|Q^\pi\|_{d,\mathrm{Lip}}$, which is typically non-identifiable purely from the data. This is because, given a sufficiently large $\eta$, we can always find a function $Q$ that satisfies all the Bellman constraints with $\|Q\|_{d,\mathrm{Lip}} \geq \eta$ by twisting with a small function; see appendix C for more details. However, if we know or can estimate the Lipschitz norm of the reward and transition functions, then it is possible to derive a theoretical upper bound of $\|Q^\pi\|_{d,\mathrm{Lip}}$ with only mild assumptions.

**Proposition 4.2.** *Let $\langle \mathcal{S} \times \mathcal{A}, d_x \rangle$ be a metric space for state action pair $x$ and $\langle \mathcal{S}, d_s \rangle$ be a metric space for state $s$. Suppose $d_x$ is separable so that $d_x(x_1, x_2) = d_s(s_1, s_2)$ if $a_1 = a_2$. If the reward function $r$ and the transition $\mathbf{T}$ are both Lipschitz in the sense that*

$$r(x_1) - r(x_2) \leq \|r\|_{\mathrm{Lip}} d_x(x_1, x_2), \quad d_s(\mathbf{T}(x_1), \mathbf{T}(x_2)) \leq \|\mathbf{T}\|_{\mathrm{Lip}} d_x(x_1, x_2), \ \forall x_1, x_2.$$

*We can prove that if $\gamma \|\mathbf{T}\|_{\mathrm{Lip}} < 1$, we have*

$$\|Q^\pi\|_{\mathrm{Lip}} \leq \|r\|_{\mathrm{Lip}}/(1 - \gamma \|\mathbf{T}\|_{\mathrm{Lip}}), \tag{15}$$

*when $\pi$ is a constant policy. Furthermore, for optimal policy $\pi^*$ with value function $Q^*$, we have:*

$$\|Q^*\|_{\mathrm{Lip}} \leq \|r\|_{\mathrm{Lip}}/(1 - \gamma \|\mathbf{T}\|_{\mathrm{Lip}}), \tag{16}$$

Theorem 4.2 suggests that if our target policy is close to the optimal, we can set $\eta = \frac{\|r\|_{\mathrm{Lip}}}{1 - \gamma \|\mathbf{T}\|_{\mathrm{Lip}}}$ if we can estimate $\|r\|_{\mathrm{Lip}}$ and $\|\mathbf{T}\|_{\mathrm{Lip}}$. This provides a way for estimating the upper bound of the Lipschitz norm of $Q^\pi$ by leveraging the Lipschitz norm for the reward and transition functions. In practice, we can estimate $\|r\|_{\mathrm{Lip}}$ and $\|\mathbf{T}\|_{\mathrm{Lip}}$ using historical data:

$$\widehat{\|r\|}_{\mathrm{Lip}} = \max_{i \neq j} \frac{(r_i - r_j)}{d(x_i, x_j)}, \qquad \widehat{\|\mathbf{T}\|}_{\mathrm{Lip}} = \max_{i \neq j} \frac{d_s(s_i', s_j')}{d_x(x_i, x_j)}. \tag{17}$$

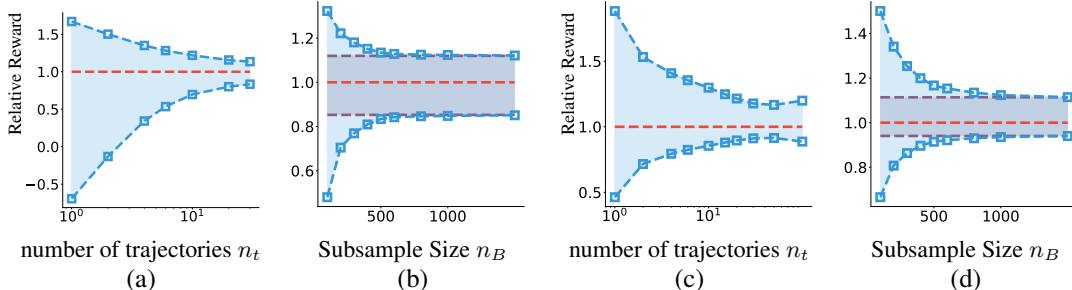

Figure 3: Results for pendulum and HIV simulator. The default settings for pendulum (Figure a,b): $n_t = 30, H = 100, \gamma = 0.95, \eta = 10.0, n_B = 500$. The default settings for HIV (Figure c,d): $n_t = 50, H = 30, \gamma = 0.75, \eta = 40.0, n_B = 500$. We follow the similar experiments from Figure 2 (b) and (c).

**Diagnosing Model Misspecification from Data**  Since the empirical maximum tends to underestimate the true maximization, simply using Proposition 4.2 may still underestimate the true Lipschitz norm. Luckily, we can diagnose if $\eta$ is too small to be consistent with data by only adding a few lines of diagnosis codes.

From Theorem 3.2 we know that for all $Q \in \mathcal{F}_\eta$, which are consistent with the finite sample Bellman equations, we have $\overline{Q}_t \succeq Q \succeq \underline{Q}_t$. Thus, if at some time $t$ (or after convergence), we find that $\overline{Q}_t(x) < \underline{Q}_t(x)$ for some $x$, we can reject the following hypothesis:

$$h : \exists Q \in \mathcal{F}_\eta, \text{ s.t. } Q(x_i) = \mathcal{B}^\pi Q(x_i), \ \forall i \in [n].$$

In this way, we can see that $Q^\pi \notin \mathcal{F}_\eta$. Then, we can increase $\eta$ by a constant factor $\kappa > 1$ and rerun our upper/lower bound algorithm. Note that we do not need to compare an infinite number of $x$ to check $\overline{Q}_t(x) < \underline{Q}_t(x)$, as $\overline{Q}_t(x) = \min_j \{ \overline{q}_{t,j} + \eta d(x, x_j) \}$ and $\underline{Q}_t(x) = \max_j \{ \underline{q}_{t,j} - \eta d(x, x_j) \}$. And hence, it is sufficient to check if there exists an index $i$ such that $\overline{q}_{t,i} < \underline{q}_{t,i}$.

## 5   Experiments

We test our algorithms in different environments. We follow Algorithm 2 with sub-sampling technique. In each environment, we evaluate the tightness of our bound by changing 1) the number of samples $n$ and 2) subsampling size $n_B$. We also make a comparison with the exact bound with full sample size (i.e. $n = n_B$). We start hyperparameter $\eta$ estimated by Proposition 4.2 with empirical maximal, and use diagnose algorithm in the last section to gradually increase $\eta$ until it is consistent with data set where we set the increasing factor $\kappa = 1.1$. We observed that diagnosis algorithm usually passes with the very initial $\eta$. As a baseline, Thomas et al. (2015b) needs huge amounts of samples to get comparable tight bounds as ours, we demonstrate a comparison experiment in Appendix D.

**Synthetic Environment with A Known Value Function**   We consider a simple environment with one dimension state space $\mathcal{S} = \mathbb{R}$ and one dimension action space $\mathcal{A} = \mathbb{R}$ with a linear transition function. Given a target policy $\pi$, we enforce our value function $Q^\pi(s, a)$ to be our predefined function. This is done by enforcing the reward function $r$ for this environment as the reverse Bellman error of $Q^\pi$, $r_i = Q^\pi(x_i) - \gamma \mathcal{P}^\pi Q^\pi(x_i)$, with $\mathcal{P}^\pi$ be the transition operator in equation (2). We use Euclidean distance metric and under this metric we can prove that the Lipschitz constant is 2. See Appendix D for more details.

All the reported results are average over 300 runs using the following setting by default: number of trajectory $n_t = 30$, Horizon length $H = 100$, discounted factor $\gamma = 0.95$, Lipschitz constant $\eta = 2.0$ and subsample size $n_B = 500$. We run Lipschitz value iteration for 100 iteration to ensure almost convergence. From Figure 2(a)(b) we can see that as the number of sample size gets larger, the bound gets tighter. 2(c) indicates that with a sufficiently large subsample size, e.g. $n_B = 500$, we can achieve bounds accurately enough compared to whole sample algorithm (purple lines). We also demonstrate the landscape of evaluation for state value function $\overline{V^\pi}(s) = \mathbb{E}_{a \sim \pi(\cdot|s)}[\overline{Q}_t(s, a)]$ and $\underline{V^\pi}(s)$ under the final Lipschitz value iteration for 100 data samples. Compare with the true value function, we can see that we get a tighter bound on a neighborhood region of data points compared to unseen region.

**Pendulum Environment**   We demonstrate our method on pendulum, which is a continuous control environment with state space of $\mathbb{R}^3$ and action space on interval $[-2, 2]$. In this environment, we aim

to control the pendulum to make it stand up as long as possible (for the large discounted case), or as fast as possible (for small discounted case). See Appendix D for more experimental setups.

Figure 3(a)(b) shows a similar result indicating our interval estimation is tight and subsampling achieves almost same tightness with full samples.

**HIV Simulator**    The HIV simulator described in Ernst et al. (2006) is a continuous state environment with 6 parameters and a discrete action environment with total 4 actions. In this environment, we seek to find an optimal drug schedule given patient's 6 key HIV indicators. The HIV simulator has richer dynamics than the previous two environments. We follow Liu et al. (2018b) to learn a target policy by fitted Q iteration and use the $\epsilon$-greedy policy of the Q-function as the behavior policy.

The default setting is similar to the previous two experiments but we use a relatively small discounted factor $\gamma = 0.75$ to ensure that we can get a reasonable Lipschitz constant from equation (15). Figure 3(c)(d) demonstrate a similar result of the HIV environment.

## 6   Conclusion

We develop a general optimization framework for off-policy interval estimation and propose a value iteration style algorithm to monotonically tighten the interval. Our Lipschitz value iteration on the continuous settings MDP enjoys nice convergence properties similar to the tabular MDP value iteration, which is worth further investigating. Future directions include leveraging our interval estimation to encourage policy exploration or offline safe policy improvement.

**Broader Impact**    Off-policy interval evaluation not only can advise end-user to deploy new policy, but can also serve as an intermediate step for latter policy optimization. Our proposed methods also fill in the gap of theoretical understanding of Markov structure in Lipschitz regression. We current work stands as a contribution to the fundamental ML methodology, and we do not foresee potential negative impacts.

**Funding Transparency Statement**    This work is supported in part by NSF CAREER #1846421, SenSE #2037267, and EAGER #2041327.

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
