[Supplementary Material]

# Appendix

## A  Lower Bound Results

We list all the results for the lower bound here.

### A.1  The Lower Bound Value Iteration

Similar to upper bound, the general algorithm for lower bound iteratively finds the lower envelope of the previous points estimation $\underline{q}_{t,i}$:

$$
\begin{aligned}
\underline{Q}_t &= \arg\min_{Q \in \mathcal{F}} \left\{ R_{\mu_{0,\pi}}[Q], \quad \text{s.t.} \quad Q(x_i) \geq \underline{q}_{t,i}, \quad \forall i \in [n] \right\} \\
\underline{q}_{t+1,i} &= \mathcal{B}^\pi \underline{Q}_t(x_i), \quad \forall i \in [n].
\end{aligned}
\tag{18}
$$

For Lipschitz functions, this yields a simple closed form solution.

**Proposition A.1.** *Suppose $\mu_{0,\pi}$ is a full-support distribution over $\mathcal{S} \times \mathcal{A}$. Consider the optimization in* (18) *with $\mathcal{F} = \mathcal{F}_\eta$ in* (8). *We have*

$$
\begin{aligned}
\underline{Q}_t(x) &= \max_{j \in [n]}(\underline{q}_{t,j} - \eta d(x, x_j)), \\
\underline{q}_{t+1,i} &= \mathcal{B}^\pi \underline{Q}_t(x_i), \quad \forall i \in [n].
\end{aligned}
\tag{19}
$$

### A.2  Convergence Results

Similar to Theorem 3.2, we have a similar monotonic result for lower bound case.

**Theorem A.2.** *Following the update in* (11) *starting from*

$$
\underline{q}_{0,i} = \frac{1}{1-\gamma} \left( r_i - \gamma\eta\mathbb{E}_{x_i' \sim \boldsymbol{T}^\pi(\cdot|x_i)}[d(x_i, x_i')] \right),
\tag{20}
$$

*we have*

$$
\underline{Q}_t \preceq \underline{Q}_{t+1} \preceq \underline{Q}^\pi, \ \forall t = 0, 1, 2, \dots,
\tag{21}
$$

*where $\underline{Q}^\pi = \arg\min_{Q \in \mathcal{F}}\{R[Q], \ \text{s.t.} \ Q(x_i) \geq \mathcal{B}^\pi Q(x_i), \ \forall i \in [n]\}$. Therefore,*

$$
R_{\mu_{0,\pi}}[\underline{Q}_t] \leq R_{\mu_{0,\pi}}[\underline{Q}_{t+1}] \leq \underline{R}_{\mathcal{F}}^\pi \leq R^\pi,
$$

*and $\lim_{t \to \infty} R_{\mu_{0,\pi}}[\underline{Q}_t] = \underline{R}_{\mathcal{F}}^\pi$.*

Similar to the linear convergence property of the upper bound case, we can establish a fast linear convergence rate for the updates in (11).

**Proposition A.3.** *Following the updates in* (11)*, we have*

$$
\underline{R}_{\mathcal{F}}^\pi - R_{\mu_{0,\pi}}[\underline{Q}_t] \leq C \frac{\gamma^t}{1-\gamma},
$$

*with constant $C := \max_{i \in [n]} |\underline{q}_{1,i} - \underline{q}_{0,i}|$.*

## B  Proofs

We focus on the proofs for upper bounds, all the lower bound proofs follows similarly.

We establish the monotonic convergence of the iterative update in section 3.2. We start with the result for general function spaces $\mathcal{F}$ and then apply it to the case of Lipschitz functions space, where $\mathcal{F}$ can ensure that the optimization in (9) is solved by a properly defined upper envelope function.

**Definition B.1.** *Given a function space $\mathcal{F}$ on domain $\Omega$ and a set of data points $(x_i, f_i)_{i=1}^n \subseteq \Omega \times \mathbb{R}$, we define the upper envelope function $g : \Omega \to \mathbb{R}$ of data points $(x_i, f_i)_{i=1}^n$ on $\mathcal{F}$ to be*

$$g(x) = \overline{\mathrm{ENV}}_{\mathcal{F}}(\{x_i, f_i\})(x) := \sup_{f \in \mathcal{F}} \{f(x) : \ s.t. \ \ f(x_i) \leq f_i, \ \ \forall i \in [n]\}, \ \forall x \in \Omega.$$

*We say that $\mathcal{F}$ is upper-self-contained if it is closed (under the infinity norm), and the upper envelope function $g : \Omega \to R$ is contained in $\mathcal{F}$ for any data points $(x_i, f_i)$ that satisfies $\inf_{f \in \mathcal{F}}\{f(x_i)\} \leq f_i, \ \forall i \in [n]$.*

*Similarly, we define the lower envelope function $g : \Omega \to \mathbb{R}$ of data points $(x_i, f_i)_{i=1}^n$ on $\mathcal{F}$ to be*

$$g(x) = \underline{\mathrm{ENV}}_{\mathcal{F}}(\{x_i, f_i\})(x) := \inf_{f \in \mathcal{F}} \{f(x) : \ s.t. \ \ f(x_i) \geq f_i, \ \ \forall i \in [n]\}, \ \forall x \in \Omega.$$

*We say that $\mathcal{F}$ is lower-self-contained if it is closed (under the infinity norm), and the lower envelope function $g : \Omega \to R$ is contained in $\mathcal{F}$ for any data points $(x_i, f_i)$ that satisfies $\sup_{f \in \mathcal{F}}\{f(x_i)\} \geq f_i, \ \forall i \in [n]$.*

Similar to contractive operator proof in value iteration, we also define the contractive property of upper/lower envelope operator $\overline{\mathrm{ENV}}_{\mathcal{F}}$ and $\underline{\mathrm{ENV}}_{\mathcal{F}}$.

**Definition B.2.** *We say $\overline{\mathrm{ENV}}_{\mathcal{F}}$ and $\underline{\mathrm{ENV}}_{\mathcal{F}}$ are contractive if for two different sets of points data $\{x_i, p_i\}_{i=1}^n$ and $\{x_i, q_i\}_{i=1}^n$, we have,*

$$\|\overline{\mathrm{ENV}}_{\mathcal{F}}(\{x_i, p_i\}_{i=1}^n) - \overline{\mathrm{ENV}}_{\mathcal{F}}(\{x_i, q_i\}_{i=1}^n)\|_\infty \leq \max_{i \in [n]} |p_i - p_i|.,$$

*and*

$$\|\underline{\mathrm{ENV}}_{\mathcal{F}}(\{x_i, p_i\}_{i=1}^n) - \underline{\mathrm{ENV}}_{\mathcal{F}}(\{x_i, q_i\}_{i=1}^n)\|_\infty \leq \max_{i \in [n]} |p_i - p_i|.$$

The following lemmas provide key properties for of this special function class. If $\mathcal{F}$ is upper-self-contained, then the optimization in (9) is solved by the upper envelop function defined above. And similarly, if $\mathcal{F}$ is lower-self-contained, then the optimization in (18) is solved by the lower envelop function defined above.

**Lemma B.3.** *If $\mathcal{F}$ is upper-self-contained/lower-self-contained, then $\overline{Q}_t$(resp. $\underline{Q}_t$) in (9)(resp. (18)) is equal to upper(resp. lower) envelope function of data points $(x_i, \overline{q}_{t,i}$(resp. $\underline{q}_{t,i})_{i=1}^n)$ almost everywhere.*

*Proof.* The upper envelope and lower envelope is inside the function space, and is maximized(resp. minimized) for all data points. Therefore they are the solutions to equation (9) and (18) almost everywhere, respectively. □

In addition, the upper and lower envelope functions is monotonic w.r.t. the data labels it goes through.

**Lemma B.4.** *In an upper-self-contained function space $\mathcal{F}$, suppose we have two sets of data points $(x_i, f_i)_{i=1}^n$ and $(x_i, g_i)_{i=1}^n$, and $\overline{f}$ and $\overline{g}$ are their upper envelope functions respectively, $\underline{f}$ and $\underline{g}$ are their lower envelopes functions respectively, if $f_i \geq g_i, \ \forall i \in [n]$, then we have $\overline{f} \succeq \overline{g}, \quad \underline{f} \succeq \underline{g}$.*

*Proof.* This is directly from the definition,

$$\overline{f}(x) = \max_{f \in \mathcal{F}} \{f(x) : \ \mathrm{s.t.} f(x_i) \leq f_i\}$$
$$\geq \max_{f \in \mathcal{F}} \{f(x) : \ \mathrm{s.t.} f(x_i) \leq g_i\}$$
$$= \overline{g}(x),$$

where the first inequality holds because the feasible region of constraints $f(x_i) \leq f_i$ is more general than $f(x_i) \leq g_i$ when $f_i \geq g_i$.

the proof for the lower envelope works similarly. □

**Lemma B.5.** *For a bounded upper-self-contained function class $\mathcal{F}$, if the upper envelope operator is contractive, then the maximum solution $\overline{Q}^\pi$ for the optimization framework equation (6) is the unique solution of the following upper-envelope Bellman equation:*

$$
\begin{aligned}
Q_i &= \mathcal{B}^\pi Q(x_i), \ \forall i \in [n], \\
Q &= \overline{\text{ENV}}(\{x_i, Q_i\}_{i=1}^n).
\end{aligned}
\tag{22}
$$

*Proof.* **Existence**

Suppose $P$ is a optimum solution for (6), if $P$ satisfies equation (22) then we are done. Otherwise $P$ satisfies $P \in \mathcal{F}$ and $P(x_i) \leq \mathcal{B}^\pi P(x_i), \ \forall i \in [n]$. Consider $q_i^\dagger = \mathcal{B}^\pi P(x_i)$, its corresponding upper-envelope function $Q^\dagger$ satisfies:

$$
Q^\dagger(x) = \max_{Q \in \mathcal{F}} \{Q(x), \ \text{s.t. } Q(x) \leq \mathcal{B}^\pi P(x_i)\} \geq P(x).
$$

Thus $Q^\dagger \succeq P$. By Bellman inequality we have:

$$
\mathcal{B}^\pi Q^\dagger(x_i) \geq \mathcal{B}^\pi P(x_i) \geq Q^\dagger(x_i),
$$

We have $Q^\dagger$ is in $\mathcal{F}$ and also satisfies Bellman inequality. By repeating this process until it converge to $Q^\infty$, we will eventually get $Q^\infty(x_i)$ satisfies equation (22) and $Q^\infty \succeq P$ which means $Q^\infty$ is also at least a optimal solution to optimization framework (6).

**Uniqueness**

Consider there are two functions $Q_1$ and $Q_2$ satisfy upper-envelope Bellman equation in (22). Consider $q_i^k = \mathcal{B}^\pi Q^k(x_i), \ \forall i \in [n], k \in \{1, 2\}$, and let $q^k$ to denote the vector of $[q_1^k, q_2^k, ..., q_n^k]^\top$, we have the infinity norm of $q^1 - q^2$ to be:

$$
\begin{aligned}
q_i^1 - q_i^2 &= \gamma \mathcal{P}^\pi (Q^1(x_i) - Q^2(x_i)) \\
&= \gamma \mathbb{E}_{x' \sim \boldsymbol{T}^\pi(\cdot|x_i)} [Q^1(x') - Q^2(x')] \\
&= \gamma \mathbb{E}_{x' \sim \boldsymbol{T}^\pi(\cdot|x_i)} [\max_{P \in \mathcal{F}} \{P(x), \ \text{s.t. } P(x_j) \leq q_j^1, \forall j \in [n]\} - \max_{P \in \mathcal{F}} \{P(x), \ \text{s.t. } P(x_j) \leq q_j^2, \forall j \in [n]\}] \\
&\leq \gamma \|q^1 - q^2\|_\infty,
\end{aligned}
$$

where the last inequality is from contractive property. This means $\|q^1 - q^2\|_\infty = 0$, and since $Q^1, Q^2$ are there corresponding upper-envelope, we have $Q^1 = Q^2$. $\qquad\square$

**Proposition 3.1 (and A.1)** *Suppose $\mu_{0,\pi}$ is a full-support distribution over $\mathcal{S} \times \mathcal{A}$. Consider the optimization in (9) with $\mathcal{F} = \mathcal{F}_\eta$ in (8). We have*

$$
\begin{aligned}
\overline{Q}_t(x) &= \min_{j \in [n]} (\overline{q}_{t,j} + \eta d(x, x_j)), \\
\overline{q}_{t+1,i} &= \mathcal{B}^\pi \overline{Q}_t(x_i), \quad \forall i \in [n]
\end{aligned}
\tag{23}
$$

*Proof.* Consider $Q \in \mathcal{F}_\eta$, for upper bound case, we have:

$$
Q(x) \leq Q(x_i) + \eta d(x, x_i) \leq \overline{q}_{t,i} + \eta d(x, x_i), \ \forall i \in [n].
$$

Therefore $Q(x) \leq \min_{i \in [n]} \{\overline{q}_{t,i} + \eta d(x, x_i)\}$.

Consider the upper envelope function $\overline{Q}_t$ which achieves $\overline{Q}_t(x) = \min_{i \in [n]} \{\overline{q}_{t,i} + \eta d(x, x_i)\}$.

By Lemma B.3 we have:

$$
\overline{Q}_t = \arg\max_{Q \in \mathcal{F}_\eta} \{R[Q], \ \text{s.t. } Q(x_i) \leq \overline{q}_{t,i}\}.
$$

Similarly we can prove for the lower bound case in Proposition A.1. $\qquad\square$

## B.1 Monotonic Convergence

It is well known that the Bellman operator is a contractive map when $\gamma \in (0,1)$, with $Q^\pi$ as the unique fixed point. Therefore, $(\mathcal{B}^\pi)^t Q$ converges to $Q^\pi$ as $t \to \infty$ for any $Q$.

Another property of special importance in our work is the monotonicity of Bellman operator. For two functions $Q_1$ and $Q_2$ on $\mathcal{S} \times \mathcal{A}$, we say that $Q_1 \succeq Q_2$ if $Q_1(s,a) \geq Q_2(s,a)$ for $\forall (s,a) \in \mathcal{S} \times \mathcal{A}$. Then we have

$$Q_1 \succeq Q_2 \quad \Rightarrow \quad \mathcal{B}^\pi Q_1 \succeq \mathcal{B}^\pi Q_2.$$

Thus if we can establish $Q \succeq \mathcal{B}^\pi Q$ (which known as the Bellman inequality (Bertsekas, 2000)), we have $Q \succeq \mathcal{B}^\pi Q \succeq (\mathcal{B}^\pi)^2 Q \succeq ... \succeq (\mathcal{B}^\pi)^\infty Q = Q^\pi$, which yields a sequence of increasingly tight upper bounds of $Q^\pi$. We leverage a similar idea to prove the monotonicity of our proposed algorithm.

We are ready to present our main result of convergence, in which we show that update (9) monotonically improves the bound and converges to the optimal solution of (6), if $\mathcal{F}$ is upper-self-contained and $\{\bar{q}_{t,i}\}$ is initialized properly such that they decrease during the first iteration.

**Theorem B.6.** *Assume $\mathcal{F}$ is upper-self-contained function set whose corresponding upper envelope operator is contractive and our evaluate distribution $\mu_{0,\pi}(s,a) = \mu_0(s) \cdot \pi(a|s)$ is full support on $\mathcal{S} \times \mathcal{A}$, e.g. $\mu_{0,\pi}(x) > 0$, $\forall x$. If we initialize the updates in (9) with $\bar{q}_{0,i}$, such that*

$$\bar{q}_{0,i} \geq \bar{q}_{1,i}, \; \forall i \in [n], \tag{24}$$

*then we have*

$$\overline{Q}_t \succeq \overline{Q}_{t+1} \succeq \overline{Q}^\pi, \; \forall t = 0, 1, 2, \ldots, \tag{25}$$

*where $\overline{Q}^\pi = \arg\max_{Q \in \mathcal{F}}\{R[Q], \text{ s.t. } Q(x_i) \leq \mathcal{B}^\pi Q(x_i), \; \forall i \in [n]\}$. Therefore,*

$$R[\overline{Q}_t] \geq R[\overline{Q}_{t+1}] \geq \overline{R_{\mathcal{F}}^\pi} \geq R^\pi,$$

*and $\lim_{t \to \infty} R[\overline{Q}_t] = \overline{R_{\mathcal{F}}^\pi}$.*

*Proof.* We focus on upper bound case, lower bound proofs follow similarly. Since $\mu_{0,\pi}$ is full support, from lemma B.3 we can see that $\overline{Q}_{t+1}$ is the upper envelope function of data points $(x_i, \bar{q}_{t+1,i})$.

Now we prove the theorem by induction on $t$ for statement $\overline{Q}_t \succeq \overline{Q}_{t+1}$.

1. Base case. $t = 0$. From Lemma B.4 we have:

$$\overline{Q}_0 \succeq \overline{Q}_1.$$

2. Induction Step. Suppose $\overline{Q}_{t-1} \succeq \overline{Q}_t$. Then $\bar{q}_{t,i} = r_i + \gamma \mathcal{P}^\pi \overline{Q}_{t-1}(x_i) \geq r_i + \gamma \mathcal{P}^\pi \overline{Q}_t(x_i) = \bar{q}_{t+1,i}$, from lemma B.4 we have:

$$\overline{Q}_t \succeq \overline{Q}_{t+1}.$$

From the induction proof we can see that $\{\overline{Q}_t(x)\}$ is a Cauchy sequence with a lower bound for every data point $x$, we know it will finally converge to a function we denote as $\overline{Q}_\infty$.

$\overline{Q}_\infty \in \mathcal{F}$ will satisfy the constraints $\overline{Q}_\infty(x_i) = \mathcal{B}^\pi \overline{Q}_\infty(x_i)$, $\forall i \in [n]$.

On the other hand, from Lemma B.5 we know that it is the $\overline{Q_{\mathcal{F}}^\pi} = Q^\infty$ almost everywhere.

This leads to a monotone sequence of measures $\{R[\overline{Q}_t]\}_{t=0}^\infty$ with a limit $\overline{R_{\mathcal{F}}^\pi}$. □

A parallel result holds for the lower bound update (11), except that the initialization condition should be $\underline{q}_{0,i} \leq \underline{q}_{1,i}$. See Appendix A for details.

**Application to Lipschitz Function Space** General convergence result can be easily applied to the case of Lipschitz functions. We show that the Lipschitz ball $\mathcal{F}_\eta$ satisfies the *upper-self-contained* condition, and provide a simple initialization method to ensure condition 24. In addition, we establish a fast convergence rate for our algorithm.

**Lemma B.7.** *i) The Lipschitz ball $\mathcal{F}_\eta = \{f : L_d(f) \leq \eta\}$ is upper-self-contained whose envelope operators are contractive.*

*ii) Following the update in* (10) *starting from*

$$\overline{q}_{0,i} = \frac{1}{1-\gamma} \left( r_i + \gamma\eta\mathbb{E}_{x_i' \sim \boldsymbol{T}^\pi(\cdot|x_i)}[d(x_i, x_i')] \right), \tag{26}$$

*we have $\overline{q}_{0,i} \geq \overline{q}_{1,i}$ for $\forall i \in [n]$.*

*Similarly, for the lower bound initializing the update in* (11) *with*

$$\underline{q}_{0,i} = \frac{1}{1-\gamma} \left( r_i - \gamma\eta\mathbb{E}_{x_i' \sim \boldsymbol{T}^\pi(\cdot|x_i)}[d(x_i, x_i')] \right),$$

*ensures $\underline{q}_{0,i} \leq \underline{q}_{1,i}$ for $\forall i \in [n]$.*

*Therefore, the results in theorem 3.2 hold.*

*Proof.* i) Consider $g(x) = \max_{f \in \mathcal{F}_\eta}\{f(x) : f(x_i) \leq f_i\}$ for given data points $(x_i, f_i)$, we can see that:

$$g(x) = \min_{i \in [n]}\{f_i + \eta d(x, x_i)\}$$

we can see that $g(x)$ is $\eta-$Lipschitz continuous.

For the contraction property, from Proposition 3.1 the upper envelope operator can be written in the following way:

$$\overline{\text{ENV}}_{\mathcal{F}_\eta}(\{x_i, f_i\}_{i \in [n]})(x) = \min_{i \in [n]}\{f_i + \eta d(x, x_i)\},$$

then we have:

$$\overline{\text{ENV}}_{\mathcal{F}_\eta}(\{x_i, f_i\}_{i \in [n]})(x) - \overline{\text{ENV}}_{\mathcal{F}_\eta}(\{x_i, g_i\}_{i \in [n]})(x)$$
$$= \min_{i \in [n]}\{f_i + \eta d(x, x_i)\} - \min_{i \in [n]}\{g_i + \eta d(x, x_i)\}$$
$$\leq \max_{i \in [n]}\{f_i - g_i\},$$

which implies contraction.

ii) For $\overline{q}_{0,i} = \frac{1}{1-\gamma}\left( r_i + \gamma\eta\mathbb{E}_{x_i' \sim \boldsymbol{T}^\pi(\cdot|x_i)}[d(x_i, x_i')]\right)$, we have:

$$\overline{q}_{0,i} = \gamma\overline{q}_{0,i} + r_i + \gamma\eta\mathbb{E}_{x_i' \sim \boldsymbol{T}^\pi(\cdot|x_i)}[d(x_i, x_i')]$$
$$= r_i + \gamma\mathbb{E}_{x_i' \sim \boldsymbol{T}^\pi(\cdot|x_i)}[\overline{q}_{0,i} + \eta d(x_i, x_i')]$$
$$\geq r_i + \gamma\mathbb{E}_{x_i' \sim \boldsymbol{T}^\pi(\cdot|x_i)}[\min_{j \in [n]}\{\overline{q}_{0,j} + \eta d(x_j, x_i')\}]$$
$$= \overline{q}_{1,i}.$$

Similarly we have $\underline{q}_{0,i} \leq \underline{q}_{1,i}$. $\qquad\qquad\square$

From Theorem B.6 and Lemma B.7 we can immediate get Theorem 3.2. Similarly we can prove the lower bound case.

## B.2 Linear Convergence

We are ready to prove the linear convergence result for Proposition 3.4 which follow the similar idea of proving linear convergence of value iteration.

**Proposition 3.3** *Following the updates in* (10), *we have*

$$R_{\mu_{0,\pi}}[\overline{Q}_t] - \overline{R_{\mathcal{F}}^\pi} \leq C\frac{\gamma^t}{1-\gamma},$$

*with constant $C := \max_{i \in [n]}|\overline{q}_{1,i} - \overline{q}_{0,i}|$.*

*Proof.* Consider $\|\overline{q}_{t+1} - \overline{q}_t\|_\infty$, where $\overline{q}_t = [\overline{q}_{t,1}, \ldots, \overline{q}_{t,n}]^\top$, we have:

$$
\begin{aligned}
\|\overline{q}_{t+1} - \overline{q}_t\|_\infty &= \max_i \{\overline{q}_{t+1,i} - \overline{q}_{t,i}\} \\
&= \gamma \max_i \mathbb{E}_{x_i' \sim \boldsymbol{T}^\pi(\cdot|x_i)} \left[ \min_{j \in [n]} \{\overline{q}_{t,j} + \eta d(x_i', x_j)\} - \min_{k \in [n]} \{\overline{q}_{t-1,k} + \eta d(x_i', x_k)\} \right] \\
&\le \gamma \max_i \mathbb{E}_{x_i' \sim \boldsymbol{T}^\pi(\cdot|x_i)} \left[ \max_{j \in [n]} \{\overline{q}_{t,j} - \overline{q}_{t-1,j} + (\eta d(x_i', x_j) - \eta d(x_i', x_j))\} \right] \\
&= \gamma \|\overline{q}_t - \overline{q}_{t-1}\|_\infty.
\end{aligned}
$$

Therefore we have $\|\overline{q}_{t+1} - \overline{q}_t\|_\infty \le \gamma^t \|\overline{q}_1 - \overline{q}_0\|_\infty$, this leads to:

$$
\begin{aligned}
\|\overline{q}_t - \overline{q}\|_\infty &= \|\overline{q}_t - \overline{q}_\infty\|_\infty \\
&\le \sum_{i=t}^\infty \|\overline{q}_{i+1} - \overline{q}_i\|_\infty \\
&\le \sum_{i=t}^\infty \gamma^i \|\overline{q}_1 - \overline{q}_0\|_\infty \\
&= \frac{\gamma^t}{1-\gamma} \|\overline{q}_1 - \overline{q}_0\|_\infty.
\end{aligned}
$$

To conclude we have:

$$
\begin{aligned}
R_{\mu_{0,\pi}}[\overline{Q}_t] - \overline{R_{\mathcal{F}}^\pi} &= \mathbb{E}_{x \sim \mu_{0,\pi}} [\overline{Q}_t(x) - \overline{Q}^\pi(x)] \\
&= \mathbb{E}_{x \sim \mu_{0,\pi}} [\min_{j \in [n]} \{\overline{q}_{t,j} + \eta d(x, x_j)\} - \min_{k \in [n]} \{\overline{q}_k + \eta d(x, x_k)\}] \\
&\le \|\overline{q}_t - \overline{q}\|_\infty \\
&= \frac{\gamma^t}{1-\gamma} \|\overline{q}_1 - \overline{q}_0\|_\infty.
\end{aligned}
$$

$\square$

## B.3 Tightness of Lipschitz-Based Bounds

**Theorem 3.4** *Let $\mathcal{F} = \mathcal{F}_\eta$ be Lipschitz function class with Lipschitz constant $\eta$. Suppose $\mathcal{X} = \mathcal{S} \times \mathcal{A}$ is a compact and bounded domain equipped with a distance $d\colon \mathcal{X} \times \mathcal{X} \to \mathbb{R}$. For a set of data points $X = \{x_i\}_{i=1}^n$, we define its covering radius to be*

$$
\varepsilon_X = \sup_{x \in \mathcal{X}} \min_i d(x, x_i).
$$

*We have*

$$
\overline{R_{\mathcal{F}}^\pi} - \underline{R_{\mathcal{F}}^\pi} \le \frac{2\eta}{1-\gamma} \varepsilon_X,
$$

*where $\eta$ is the Lipschitz constant and $\gamma$ the discount factor.*

*Proof.* Consider $\|\overline{q} - \underline{q}\|_\infty$, where $\overline{q} = [\overline{q}_1, \overline{q}_2, \ldots, \overline{q}_n]^\top$ and $\underline{q} = [\underline{q}_1, \underline{q}_2, \ldots, \underline{q}_n]^\top$ as the vector of upper and lower functions $\overline{Q}, \underline{Q}$ at point $x_1, x_2, \ldots, x_n$.

Since $r_i = q_i - \gamma \mathcal{P}^\pi Q(x_i)$ for all $Q$ satisfies the finite points Bellman equation (see more details in Lemma B.5), we have:

$$
\begin{aligned}
\overline{q}_i - \underline{q}_i &= \gamma \mathcal{P}^\pi (\overline{Q}(x_i) - \underline{Q}(x_i)) \\
&= \gamma \mathbb{E}_{x_i' \sim \boldsymbol{T}^\pi(\cdot|x_i)} \left[ \overline{Q}(x_i') - \underline{Q}(x_i') \right] \\
&\le \gamma \mathbb{E}_{x_i' \sim \boldsymbol{T}^\pi(\cdot|x_i)} \left[ \min_j \{\overline{q}_j - \underline{q}_j + 2\eta d(x_i', x_j)\} \right] \\
&\le \gamma \|\overline{q} - \underline{q}\|_\infty + 2\eta \gamma \mathbb{E}_{x_i' \sim \boldsymbol{T}^\pi(\cdot|x_i)} \left[ \min_j d(x_i', x_j) \right].
\end{aligned}
$$

Let $\varepsilon_n = \sup_{x \in \mathcal{X}} \min_i \{d(x, x_i)\}$, which is the epsilon ball radius given centers $\{x_i\}_{i=1}^n$, typically $\varepsilon \approx O(n^{-\frac{1}{d}})$. We have:

$$\|\overline{q} - \underline{q}\|_\infty = \max_i |\overline{q}_i - \underline{q}_i|$$

$$\leq \gamma \|\overline{q} - \underline{q}\|_\infty + 2\eta\gamma \max_i \mathbb{E}_{x_i' \sim \boldsymbol{T}^\pi(\cdot|x_i)} \left[ \min_j d(x_i', x_j) \right]$$

$$\leq \gamma \|\overline{q} - \underline{q}\|_\infty + 2\eta\gamma \sup_{x \in \mathcal{X}} \min_j d(x, x_j)$$

$$= \gamma \|\overline{q} - \underline{q}\|_\infty + 2\eta\gamma\varepsilon_n.$$

Thus we have:

$$\|\overline{q} - \underline{q}\|_\infty \leq \frac{2\eta\gamma}{1-\gamma}\varepsilon_n$$

Consider $\|\overline{Q} - \underline{Q}\|_\infty$, we have:

$$\|\overline{Q} - \underline{Q}\|_\infty = \sup_{x \in \mathcal{X}} |\overline{Q}(x) - \underline{Q}(x)|$$

$$\leq \sup_{x \in \mathcal{X}} |\min_j \{\overline{q}_j - \underline{q}_j + 2\eta d(x, x_j)\}|$$

$$\leq \|\overline{q} - \underline{q}\|_\infty + 2\eta \sup_{x \in \mathcal{X}} \min_j \{d(x, x_j)\}$$

$$\leq \frac{2\eta\gamma}{1-\gamma}\varepsilon_n + 2\eta\varepsilon_n$$

$$= \frac{2\eta}{1-\gamma}\varepsilon_n.$$

Therefore we have:

$$\overline{R_{\mathcal{F}}^\pi} - \underline{R_{\mathcal{F}}^\pi} = \mathbb{E}_{x \sim \mu_{0,\pi}}[\overline{Q}(x) - \underline{Q}(x)]$$

$$\leq \frac{2\eta}{1-\gamma}\varepsilon_n$$

$\square$

## B.4   Additional Proofs

**Proposition 4.1**   *Follow equation* (14) *with initialization follows* (12), *let* $\overline{Q}_t$ *to be the upper envelope function of data points* $\{x_i, \overline{q}_{t,i}\}_{i=1}^n$, *we have a similar monotonic result as theorem 3.2:*

$$\overline{Q}_t \succeq \overline{Q}_{t+1} \succeq \overline{Q}^\pi, \ \forall t = 0, 1, 2, \ldots,$$

*Proof.* The proof is the same as Theorem B.6, once we see that we gradually $\overline{q}_i$ but they will always stay ahead the upper bound $\overline{Q}^\pi(x_i)$, but the convergence is the same as the original algorithm. $\square$

**Proof of Theorem 4.2**   *Let* $\langle \mathcal{S} \times \mathcal{A}, d_x \rangle$ *be a metric space for state action pair* $x$ *and* $\langle \mathcal{S}, d_s \rangle$ *be a metric space for state* $s$. *Suppose* $d_x$ *is separable so that* $d_x(x_1, x_2) = d_s(s_1, s_2)$ *if* $a_1 = a_2$. *If the reward function* $r$ *and the transition* $\boldsymbol{T}$ *are both Lipschitz in the sense that*

$$r(x_1) - r(x_2) \leq \|r\|_{\mathrm{Lip}} d_x(x_1, x_2)$$
$$d_s(\boldsymbol{T}(x_1), \boldsymbol{T}(x_2)) \leq \|\boldsymbol{T}\|_{\mathrm{Lip}} d_x(x_1, x_2), \ \forall x_1, x_2.$$

*We can prove that if* $\gamma \|\boldsymbol{T}\|_{\mathrm{Lip}} < 1$, *we have*

$$\|Q^\pi\|_{\mathrm{Lip}} \leq \frac{\|r\|_{\mathrm{Lip}}}{1 - \gamma \|\boldsymbol{T}\|_{\mathrm{Lip}}}, \tag{27}$$

*when* $\pi$ *is a constant policy. Furthermore, for optimal policy* $\pi^*$ *with value function* $Q^*$, *we have:*

$$\|Q^*\|_{\mathrm{Lip}} \leq \frac{\|r\|_{\mathrm{Lip}}}{1 - \gamma \|\boldsymbol{T}\|_{\mathrm{Lip}}}, \tag{28}$$

*Proof.* Suppose $a_i = \arg\max_a\{Q^\pi(\boldsymbol{T}^i(x_1), a) - Q^\pi(\boldsymbol{T}^i(x_2), a)\}$, where $\boldsymbol{T}^1(x_j) = \boldsymbol{T}(x_j)$ and $\boldsymbol{T}^i(x_j) = \boldsymbol{T}(\boldsymbol{T}^{i-1}(x), a_{i-1})$, $\forall j \in \{1, 2\}$ is defined recursively.

If $\pi$ is a constant policy where $\pi(a|s_1) = \pi(a|s_2) = \pi(a)$, $\forall a, s_1, s_2$, we can actually write $Q^\pi(x) - Q^\pi(x_2)$ as:

$$Q^\pi(x_1) - Q^\pi(x_2) = (r(x_1) - r(x_2)) + \gamma \int_a \left(\pi(a|T(x_1))Q^\pi(\boldsymbol{T}(x_1), a) - \pi(a|T(x_2))Q^\pi(\boldsymbol{T}(x_2), a)\right) da$$

$$= (r(x_1) - r(x_2)) + \gamma \int_a \pi(a) \left(Q^\pi(\boldsymbol{T}(x_1), a) - Q^\pi(\boldsymbol{T}(x_2), a)\right) da$$

$$\leq (r(x_1) - r(x_2)) + \gamma \max_a \left(Q^\pi(\boldsymbol{T}(x_1), a) - Q^\pi(\boldsymbol{T}(x_2), a)\right)$$

$$= (r(x_1) - r(x_2)) + \gamma \left(Q^\pi(\boldsymbol{T}(x_1), a_1) - Q^\pi(\boldsymbol{T}(x_2), a_1)\right).$$

And similarly we have

$$Q^\pi(\boldsymbol{T}^{i-1}(x_1), a_{i-1}) - Q^\pi(\boldsymbol{T}^{i-1}(x_2), a_{i-1}) \leq r(\boldsymbol{T}^{i-1}(x_1), a_{i-1}) - r((\boldsymbol{T}^{i-1}(x_2), a_{i-1})) + \gamma \left(Q^\pi(\boldsymbol{T}^i(x_1), a_i) - Q^\pi(\boldsymbol{T}^i(x_2), a\right.$$

Therefore we have:

$$Q^\pi(x_1) - Q^\pi(x_2) \leq (r(x_1) - r(x_2)) + \sum_{i=1}^\infty \gamma^i (r(\boldsymbol{T}^i(x_1), a_i) - r(\boldsymbol{T}^i(x_2), a_i))$$

$$\leq \lambda_r d_x(x_1, x_2) + \sum_{i=1}^\infty \gamma^i (r(\boldsymbol{T}^i(x_1), a_i) - r(\boldsymbol{T}^i(x_2), a_i)) \text{ //according to definition of max operator over } a_i$$

$$\leq \lambda_r d_x(x_1, x_2) + \sum_{i=1}^\infty \gamma^i \lambda_r d_x([\boldsymbol{T}^i(x_1), a_i], [\boldsymbol{T}^i(x_2), a_i]) \text{ //Lipschitz of reward function}$$

$$= \lambda_r d_x(x_1, x_2) + \sum_{i=1}^\infty \gamma^i \lambda_r d_s(\boldsymbol{T}^i(x_1), \boldsymbol{T}^i(x_2)) \text{ //by the assumption } d_x(x_1, x_2) = d_s(s_1, s_2) \text{ if } a_1 = a_2$$

$$\leq \lambda_r \left(d_x(x_1, x_2) + \sum_{i=1}^\infty \gamma^i \lambda_T^i d_x(x_1, x_2)\right)$$

$$= \frac{\lambda_r}{1 - \gamma\lambda_T} d_x(x, \bar{x}).$$

The last inequality can be proved inductively by

$$d_s(\boldsymbol{T}^i(x_1), \boldsymbol{T}^i(x_2)) \leq \lambda_T d_x([\boldsymbol{T}^{i-1}(x_1), a_{i-1}], [\boldsymbol{T}^{i-1}(x_2), a_{i-1}])$$

$$= \lambda_T d_s(\boldsymbol{T}^{i-1}(x_1), \boldsymbol{T}^{i-1}(x_2)$$

$$\leq ...$$

$$\leq \lambda_T^{i-1} d_s(\boldsymbol{T}(x_1), \boldsymbol{T}(x_2))$$

$$\leq \lambda_T^i d_x(x_1, x_2).$$

For $Q^*$ case we can have the similar derivation where at the beginning of the proof we have:

$$Q^*(x_1) - Q^*(x_2) = (r(x_1) - r(x_2)) + \gamma(\max_a\{Q^\pi(\boldsymbol{T}(x_1), a)\} - \max_a\{Q^\pi(\boldsymbol{T}(x_2), a)\})$$

$$\leq (r(x_1) - r(x_2)) + \gamma \max_a \left(Q^\pi(\boldsymbol{T}(x_1), a) - Q^\pi(\boldsymbol{T}(x_2), a)\right)$$

$\square$

## C More Discussions on Lipschitz Norm

We show the non-identifiable results of upper bound of Lipschitz norm by constructing a set of possible $Q$ functions that are consistent with the data but with an unbounded increasing Lipschitz norm.

We first show that if our function set provides at least two different solutions, we have a nontrivial solution in null space.

**Lemma C.1** (Null Space of Finite Bellman Constraints). *There is a non-zero Lipschitz continual function $G$ with $\|G\|_{d,\text{Lip}} \leq 2\eta$ such that:*

$$G(x_i) = \gamma \mathcal{P}^\pi G(x_i), \ \forall i \in [n],$$

*once the solution in $\mathcal{F}_\eta$ is not unique.*

*Proof.* Suppose $Q_1, Q_2 \in \mathcal{F}_\eta$ satisfies all the finite Bellman constraints. Consider $G = Q_1 - Q_2$, we have $\|G\|_{d,\text{Lip}} \leq 2\eta$ and

$$G(x_i) = Q_1(x_i) - Q_2(x_i) = \gamma \mathcal{P}^\pi (Q_1 - Q_2)(x_i) = \gamma \mathcal{P}^\pi G(x_i).$$

□

Using the nontrivial solution in null space we can construct arbitrarily large Lipschitz norm solution of $Q$ that are consistent with data.

**Theorem C.2** (Non-identifiable of Upper Bound of Lipschitz Function). *If there is more than one Lipschitz functions satisfies finite Bellman constraints, For all $\eta$ we can always find $Q$ satisfies Bellman constraint and $\|Q\|_{d,\text{Lip}} \geq \eta$.*

*Proof.* By using $G$ in Lemma C.1 we can construct a set of $Q_\lambda$ satisfies finite samples Bellman equation in (4).

$$Q_\lambda = Q^\pi + \lambda G.$$

where if we pick $\lambda \geq (\eta + \|Q^\pi\|_{d,\text{Lip}})/\|G\|_{d,\text{Lip}}$ we have:

$$\|Q_\lambda\|_{d,\text{Lip}} \geq (\eta + \|Q^\pi\|_{d,\text{Lip}}) - \|Q^\pi\|_{d,\text{Lip}} = \eta.$$

and $Q_\lambda$ satisfies finite samples Bellman constraints:

$$Q_\lambda(x_i) = Q^\pi(x_i) + \lambda G(x_i) = r_i + \gamma \mathcal{P}^\pi (Q^\pi(x_i) + \lambda G(x_i)) = \mathcal{B}^\pi Q_\lambda(x_i).$$

□

# D   Experimental Settings

**Comparison Results using Thomas BoundsThomas et al. (2015b)**   We compare our method with lower bound estimation from Thomas et al. (2015b), whose bound leverage a sophisticated concentration bound by importance sampling estimators. Their method is based on the unbiased *importance sampling* (Precup et al., 2000) estimator of $\mathcal{R}^\pi$:

$$\hat{R}^\pi_{\text{IS}} := \underbrace{R(\tau)}_{\text{return}} \underbrace{\prod_{t=1}^T \frac{\pi(a_t|s_t)}{\pi_0(a_t|s_t)}}_{\text{importance weight}},$$

where $\tau$ is the trajectory and $R(\tau)$ is the normalized and discounted average reward of a trajectory. They obtain a high confidence lower bounds for $\hat{R}^\pi_{\text{IS}}$ by leveraging the concentration inequality with an adjust threshold parameter specified by user.

| Number of Trajectories | 2 | 4 | 6 | 10 | 20 | 30 |
|---|---|---|---|---|---|---|
| Thomas Relative Lower Bound | **-8.61e-03** | -2.87e-03 | -1.72e-03 | -9.56e-04 | -4.53e-04 | -2.97e-04 |
| Our Relative Lower Bound | -0.131 | **0.343** | **0.536** | **0.698** | **0.805** | **0.850** |

Table 1: Comparison with Thomas Lower Bound Thomas et al. (2015b), close to 1 is better.

We empirically evaluate the method on Pendulum environment, where we use the same default settings as we conduct experiments for our algorithms. Table 1 shows the results compared to our lower

Figure 4: Landscape of upper and lower bound of $V^\pi$ with different number of samples.

bound. We pick the best result choosing the threshold number from $\{0.001, 0.01, 0.1, 1.0, 10.0\}$ and we set the confidence level of Thomas lower bound to be $95\%$. All the numbers are the relative reward that divided by the ground truth.

We can see that Thomas' lower bound is not sensitive to small number of samples, which is almost near 0. This is mainly because the importance ratio between the target policy $\pi(a|s)$ and the behavior policy $\pi_0(a|s)$ for each trajectory sample goes to 0 due to the curse of horizon (we use horizon length = 100 here) , which makes IS based estimator not a proper method for long (or infinite) horizon problems. As a consequence, concentration confidence bounds based on IS estimators could be potentially loose in such problems (The number of trajectories used in Thomas et al. (2015b) can be $n = 10^7$ if they want to get a tight lower bound).

**Synthetic Environment with A Known Value Function**    The transition of this environment is a one dimension linear function:

$$\boldsymbol{T}(s, a) = 0.8s - 0.4a - 0.1,$$

and the target policy we use is

$$\pi(s, \xi) = 1.5s - 0.1 + \xi, \ \xi \sim \mathcal{N}(0, \sigma^2),$$

where Gaussian variance $\sigma = e^{-5}$. And the historical data is pre-collected by a behavior policy $\pi_0$ similar to $\pi$ but with a larger variance.

The predefined q-function is $Q^\pi(s, a) = f(s) + f(a - \frac{\pi}{2})$ where $f(x) = \sqrt{x^2 + x\sin(x) + 1}$. For distance metric we use Euclidean distance $d(x, y) = \|x - y\|_2$. Under this distance metric, we can calculate the exact Lipschitz constant:

$$
\begin{aligned}
L_d(Q^\pi) &= \sup_{s,a} \lim_{\epsilon \to 0} \frac{Q^\pi(s, a) - Q^\pi(s + \frac{\sqrt{2}}{2}\epsilon, a + \frac{\sqrt{2}}{2}\epsilon))}{\epsilon} \\
&= 2 \sup_s \lim_{\epsilon \to 0} \frac{f(s) - f(s + \frac{\sqrt{2}}{2}\epsilon)}{\epsilon} \\
&= 2
\end{aligned}
$$

Figure 4 shows a full landscape of upper and lower bounds of state value function $V^\pi$ under different number of samples, similar to Figure 2(e).

**Pendulum Environment**    We learn a feature map $\Phi$ for $Q^\pi$ by a two hidden layers neural network $[f_1, f_2, f_3]$, where the input layer is state action pair $x_0 = x$, the first hidden layer is $x_1 = f_1(x) = \text{RELU}(W_1^\top x + b_1)$ with 100 hidden dimension matrix $W_1, b_1$. We set $x_2 = f_2(x_0, x_1) = [x_0, \text{RELU}(W_2^\top x_1 + b_2)]^\top$ as feature layer, where we let the concatenate the input layer and a relu of linear layer for the hidden layer as our feature. We add the input layer to our feature layer to ensure that our distance function $d(x_0, \bar{x}_0) = \|x_2 - \bar{x}_2\|$ is a true distance function (not a semi-distance one because $x_2 = \bar{x}_2$ requires $x = \bar{x}$). And finally the last layer is a linear layer with output dimension 1 as the output of q-function. Thus our approximate q-function can be represented as

$$Q(x) = W_3^\top \Phi(x) + b_3,$$

where $\Phi(x) = f_2(x, f_1(x))$.

We apply fitted value evaluation algorithm Munos & Szepesvári (2008); Le et al. (2019) to use off-policy data to pre-train a $Q^\pi$, then we use the feature layer $x_2 = \Phi(x)$ as our feature, and set the Lipschitz constant approximately 2 times the $\ell_2$ norm of the last linear layer parameter $W_3$ as our default parameter.

**HIV simulator** We follow exactly the same settings as Liu et al. (2018b).