[Reviews · NeurIPS 2020]

Review 1

Summary and Contributions: The authors present an algorithm for providing an upper and lower bound on the value of a policy based on arbitrary transition samples, in the setting where the rewards and transitions are deterministic but the policy is random. The bounds are derived by considering a ball of Lipschitz Q functions and assumes that the true Q function exists in the proposed set. Experiments are presented in three domains.

Strengths: - Interesting viewpoint on bounds in deterministic MDP settings, clear progression from theory to practice - Provides a practical algorithm

Weaknesses: - Only addresses deterministic settings (not a significant weakness; this was not the point of the paper)

Correctness: I believe they are correct

Clarity: There are several issues with clarity; see comments below.

Relation to Prior Work: It's important to clearly describe the setting of this work. For example, in the paragraph on l.48, limitations of related work are mentioned, like for example a "heavy reliance on i.i.d. data from a particular behaviour policy." I think most (all?) of the referenced work provide confidence intervals in settings where the transitions and/or rewards are not deterministic. So they have to deal with uncertainty in reward and transition estimates, depending on the framework. This work, on the other hand, assumes deterministic rewards and transitions, "for simplicity," so as soon as a (s,a,s,r) pair is observed, that part of the reward and transition functions is completely characterized, and it doesn't matter how it was collected. This is fine, but it takes away the very problem that the cited works are trying to address. So I find the criticisms of other work in the introduction to be unhelpful, or at worst, misleading. In particular, a reader who is not familiar with the field could be easily misled by what is written. This paper is interesting and doesn't have to position itself as "better" than work that is addressing a very different setting with different concerns. Rather it should describe how it is addressing different concerns by making different assumptions. This occurs again at line 118, where the authors note that "unlike standard concentration bounds..." but the need for those kinds of bounds has been removed because there is no need for estimation in the assumed setting.

Reproducibility: Yes

Additional Feedback: Throughout the word "reward" is used in a way that's not precise; I think it usually refers to "expected reward" in a way that would usually mean the "value of a policy" (under some initial state distribution.) l.28 - "trustful" - trustworthy l.72 - "we define the infinite horizon discounted reward" - *expected* reward l.77 - "Our method requires the least assumptions on the off-policy data." - Relative to what? What does this mean? Clarify. l.78 - At this point, explain briefly the properties you want the interval estimator to have. l.87 - second s_i is missing a prime. Also \gamma is missing? l.91 - What about the situation where there is no solution in F? l.102 - Does this relation hold pointwise or uniformly? l.111 - "worthy to note" - worth noting l.114 - "as data points increase" - they have to increase in a particular way (covering S x A) - make more precise Alg 1 - "discounted factor" - discount factor Figure 1 has no axis labels and no caption. l.181 - scarify - I do not think this means what you think it means. Section 5: Give an idea of how long computation takes. ===== I thank the authors for their response. I just want to note that in the reviewer discussion, we noticed that in B.1 there was some confusion among the reviewers on when Bpi meant to indicate the "true" Bellman operator, and when it meant to indicate the version learned from/restricted to the data. While it may not matter which one the reader is thinking of at that point, it may be worth clarifying this if possible. (This may have implications in the main paper also.)


Review 2

Summary and Contributions: Authors tackled OPPE problem, and they proposed a method to quantify upper and lower bound on the value of the evaluation policy using Lipschitz constraint on the function class. The main benefit of their method is 1. no requirement on i.i.d samples because they are not using concentration inequalities 2. tighter bounds, and the main challenges are 1. computation (for solving the optimization problem, even the closed form is more computationally demanding than for example Thomas et al.), 2. validating the right Lipschitz constraint (although authors tried to address it). Authors additionally evaluated their methods with in experimental setting.

Strengths: There are some great points about the paper, 1. Writing, I found it easy to follow and the right balance of putting details into the main text and the appendix. 2. Their commitment to have a an applicable algorithm, they addressed the main two issue that may arise from moving away from merely theoretical perspective: computation time (with sub sampling) and hyper-parameter (which they addressed by checking \eta).

Weaknesses: Empirical Evaluation, I believe authors could have done a better job in comparing their algorithm to existing approaches, these are included but not limited to Bootstrapping and different concentration inequalities. (A really nice source is Phil Thomas's thesis (Safe Reinforcement Learning), where he compares different concentration inequalities, t-student distribution and bootstrapping). For example BCa (Bias Corrected Accelerated Bootstrapping) seems to show a better performance in low data regime (also I'm not sure which bound of Thomas et al they used int their comparison). That being said, I believe the contributions of the paper is significant enough and I won't suggest a rejection based on lack of enough experiments (happy to hear other reviewers perspective)

Correctness: To the best of my knowledge, I believe the proofs are correct.

Clarity: yes. However, 1. using R[Q] as expectation is very confusing notation, I suggest authors revisit this choice of notation. 2. Q >= \tilde{Q} .. in line 105 : this has not been defined, what does it exactly mean for a function to be greater? (My guess is that this should be true for all "x")

Relation to Prior Work: Good. Line 55 : "which mainly focus on the regret bound or sample complexity" : PAC-RL is different than regret bound, and saying PAC-RL is mainly focused on regret bound is technically a wrong statement, I urge the authors to change this. Double citation : Thomas, P. S., Theocharous, G., and Ghavamzadeh, M. High-confidence off-policy evaluation. In Twenty-Ninth AAAI Conference on Artificial Intelligence, 2015b.

Reproducibility: Yes

Additional Feedback: Additionally I wonder 1. What are authors thought about using other forms of constraints rather than Lipschitz? Also, I have not found in the paper authors justifying why this is a good assumption? Is there an example of a value function that does not satisfy this assumption? 2. What is the effect of distance function? and why authors think L2 is a good choice? 3. How does this method perform in larger state space? I believe HIV is in R^6, any intuition about what happens if we go to larger dimensions, for example ICU data from MIMIC can go up to 50 or more dimension. ## Thanks for providing feedback, I read the feedback and will keep my score as it was.


Review 3

Summary and Contributions: RE authors' feedback: I understand that the extension to the stochastic setting is beyond the scope of this work, but it would be great to add the gist of the idea (as mentioned in the authors rebuttal) to the paper or appendix so that the reader knows about potential extension path. RE authors' feedback: Check out the literature on "order statistics" for references on estimating distribution min/max under distributional assumptions. This paper presents a method for provably correct interval estimation in off-policy evaluation with continuous state and action RL domains. The main idea is to find the smallest and largest expected reward that is consistent with the observed data under a Lipschitz continuity assumption of the Q function. The data is assumed to be collected under a blackbox and unknown behavior policy, thus avoiding the typical i.i.d. sampling assumption. The paper solves the constraint optimization problem by introducing a Bellman-like update rule with similar contraction properties as normal Bellman update. This update is applied in several iterations on the upper/lower estimates of the Q value for the observed points. It requires sampling actions from the target policy and finding the upper/lower bound by iterating over the observed data, resulting in a runtime quadratic in the sample size. The paper suggests an approximation (still resulting in theoretically valid but less-tight bounds) that uses a subset of observed transitions in each iteration of the Bellman update to reduce the complexity of the algorithm. To set the Lipschitz constant required by the algorithm, the paper suggests a method for estimating the Lipschitz constant of the Q functions based on the constants for the reward and transition functions. The empirical results are minimal, but IMO enough to establish the relevance and usefulness of the proposed algorithm.

Strengths: - If the Lipschitz assumption holds for the true Q function, the resulting intervals are theoretically guaranteed to be correct. The assumptions used in the algorithm are relatively mild for continuous state/action spaces. - The algorithm only has a couple of parameters to tune/estimate (Lipschitz constant). Since the Bellman-like update rules have monotonic and linear convergence (a contraction), the algorithm converges in a few iterations. - The algorithm is fairly simple and easy to implement.

Weaknesses: - Rewards and transitions are assumed to be deterministic. The paper does not explain if any of the proposed results can be extended to the stochastic case. IMO much of this work is not possible to extend to the stochastic transition case without requiring non-trivial modeling assumptions. - The quadratic dependence on the sample size is problematic and random sampling in each iteration can only go so far (e.g. in the experiments, it required using ~1/6 of the data) to get close to the tight optimal bound. - Requires knowledge of the Lipschitz constant for a tight bound.

Correctness: I did not fully check the proofs in the appendix. The theoretical results look very reasonable and correct.

Clarity: The paper is clearly written and easy to follow.

Relation to Prior Work: Most of the relevant work either require strong assumptions or are only applicable in tabular or linear function spaces.

Reproducibility: Yes

Additional Feedback: There is a body of literature on estimating distribution min/max form sample min/max with mild distributional assumptions. Such techniques can be used to get tighter estimates of the Bellman update with subsampling in (14), and also in estimating the Lipschitz constant for rewards and transition functions.


Review 4

Summary and Contributions: The authors propose an algorithm for obtaining upper and lower bounds on the value function in off-policy reinforcement learning. Specifically, they propose a fitted value iteration algorithm that uses Lipschitz continuous functions as the model family, and prove that their algorithm converges. They provide a practical variant of their algorithm, and experimentally validate it on several benchmarks.

Strengths: Important problem Interesting technical contribution

Weaknesses: Some assumptions that appear strong (in particular, deterministic dynamics)

Correctness: Yes, the claims appear to be correct.

Clarity: The paper is decently well written.

Relation to Prior Work: Yes, the prior work is discussed.

Reproducibility: Yes

Additional Feedback: My most important concern is that the authors assume that the transitions and rewards are deterministic, saying this is “for simplicity”. However, this appears to me to be a rather strong assumption; for high-dimensional state spaces, exponentially many samples may be required for estimates to converge. The authors need to clarify how their approach would be extended to stochastic dynamics. This is especially important since they cite domains such as healthcare as applications, where stochasticity is prevalent. The sample complexity (to upper bound the gap between the upper and lower bounds) given in Theorem 3.4 appears to be exponential in the effective dimension (which is expected since the model family is nonparametric). If my understanding is correct, then I think the authors should be more upfront about this dependence. The empirical results look promising even on the HIV simulator with a 6 dimensional state space, but an evaluation of the dependence of the bounds on the state space dimension using the synthetic data would have been helpful for understanding this tradeoff. Theorem 3.2 is a bit concerning -- does it only hold for a specific initial point? I might have expected a result along the lines of “for all initial points \bar{q}_{0,i} that are at least this value”, but a single initial point seems overly restrictive. The notation in the paper is confusing at times, and I do not believe it is standard. Using more standard notation would make the paper more readable (but less concise). As a minor suggestion, it would be helpful to emphasize that the state space is continuous and that the action space can be continuous in the background section. On line 87, I assume the authors mean tuples (s_i, a_i, s_i’, r_i) (the prime is on r_i rather than the second s_i). ----------------------------------------------------------------------------------------- Post-rebuttal: Thank you for the clarifications; I will keep my score as-is.

[Author Response · NeurIPS 2020]

**[General Response]** We thank all reviewers for the detailed and valuable feedback. We will fix the typos and improve
the draft carefully based on your comments.

*Q1.Limitation of the deterministic setting and extension to stochastic settings*: Our main contribution is designing
a simple value iteration style algorithm to get the interval estimation. For the sake of clarity, we choose to focus on
the deterministic transition and reward settings. One way to extend our results to stochastic settings is to follow the
Appendix C proof's technique in [1], by decomposing the empirical operator $\hat{\mathcal{B}}^\pi$ as $\mathcal{B}^\pi + (\hat{\mathcal{B}}^\pi - \mathcal{B}^\pi)$, where we can
bound the latter operator $(\hat{\mathcal{B}}^\pi - \mathcal{B}^\pi)$ via Rademacher complexity of the Lipschitz function class. To avoid digressing
from the primary focus of the current paper, we decide to leave them as our future work.

**[Reviewer #1]** We thank reviewer #1 for your valuable suggestions and detailed writing corrections.

*Q2.Misleading claim on non i.i.d. assumption*: Even when assuming deterministic transitions, typical approaches
based on concentration inequalities still require i.i.d. conditions on the transition $(s_i, a_i)$ pairs, or can only work on the
trajectory level as in [2] (which has a smaller effective sample size). We plan to have a new section to compare the
concentration inequality approach with our method side by side to further clarify the raised concerns.

**[Reviewer #2]** We thank reviewer #2 for your insightful questions and valuable experimental references.

*Q3.Empirical comparison to existing approaches*: We have a comparison with [2] in Table 1 of Appendix D. In general,
[2] views each trajectory as one sample while we view each transition pair as one sample. The result in Appendix
D shows that our method is better than that of [2] when the number of trajectories is small. Moreover, like other
trajectory-based IS methods, [2] also suffers from the curse of horizon. We will add more discussion in the revision.

*Q4. Reason to choose Lipschitz function class*: We had a brief discussion on this in line 128-129. We choose the
Lipschitz function class to make a good balance between expressiveness and tightness of the bounds. More specifically,
it includes a very rich set of functions that could cover the true value function with high probability, while allowing us
to get practical bounds with a simple algorithm. We will add more discussion in the final draft.

*Q5.Distance function and high dimensional state space*: We find out $L_2$ distance is enough for the low-dimension
environment. In high-dimension data, we may need to find a better distance measure to capture the underlying
low-dimension manifold of the data, which seems to be an exciting direction to explore. We will leave it as future work.

**[Reviewer #3]** We thank reviewer #3 for your valuable comments and suggestions.

*Q6.Quadratic dependency on sample size and random sample method seems not ideal*: We avoid the quadratic
dependency by adopting the random sub-sampling technique, which may sacrifice the tightness of the interval bounds
for reducing the computation burden. Moreover, as the sub-sampling bounds are still provably correct bounds of the
true $R^\pi$ (despite being less tight), the sub-sampling interval bounds can still guide the end-user for decision-making. In
real world applications, we can trade off the tightness and the computation complexity conditioning on the available
computation resource.

*Q7.Require knowledge of Lipschitz constant*: We agree that Lipschitz constant is crucial for the success of a valid
interval estimation. We emphasize and discuss it in section 4.2.

*Q8.Related literature on estimating distribution min/max form which can improve equation (14)*: Thanks for pointing
out the reference. It sounds very interesting! Could you kindly send the references in the revised rebuttal?

**[Reviewer #4]** We thank reviewer #4 for your valuable comments and suggestions.

*Q9.The sample complexity appears to be exponential in the effective dimension*: We agree that the sample complexity
is exponential and the main reason to choose Lipschitz function class is because it strikes a good balance between
richness and simplicity. In line 180 we pointed out that: "it is possible to choose smaller space sets (such as RKHS) to
obtain smaller gaps, it would scarify other properties such as capacity and simplicity."

*Q10.Theorem 3.2 for the specified initial points*: We can start with an arbitrary initial point and still achieve linear
convergence as in Proposition 3.3, which means that when the algorithm converges we will get a pair of provably
correct bounds. However, with the specified initial point in Theorem 3.2, we can have a stronger guarantee on **anytime**
bounds, which means that whenever we stop the algorithm (before it converges), the upper and lower bounds we get is
guaranteed to include $R^\pi$. Moreover, we believe that the calculation of the initial point is not difficult (see Eq (12)).

**[References]**

[1] Mousavi, Li, Liu, Zhou. Black-box Off-policy Estimation for Infinite-Horizon Reinforcement Learning, ICLR 2020.

[2] Thomas, Theocharous, Ghavamzadeh. High-confidence off-policy evaluation. AAAI 2015


[Meta-Review · NeurIPS 2020]

This paper provides a novel strategy to get upper and lower bounds on action-values, under assumptions that the action-values are Lipschitz and that the environment is deterministic. The paper has a novel idea, but could benefit from a few key changes to significantly improve the paper. 1. Reviewers pointed out making less strong claims about issues with previous approaches, since determinism can be exploited here. In the response, you said you would contrast this more clearly, and I highly recommend spending a reasonable amount of time explaining how this new approach you are taking differs from the more common (and well-understood) statistical approaches like bootstrapping. 2. The restriction to deterministic problems is acceptable. But, a discussion about extensions to the stochastic setting would make the contribution stronger. 3. Some of the notation needs to be improved. As mentioned by a reviewer, R[Q] is odd. We also got confused in some places by interchanging Bpi as meaning the true Bellman operator and the one based only on observed data. This operator should not be overloaded, and it might be better to call the one based on data Bhatpi, or even Bpi_n where n is the number of samples.